# Accelerating Model-Based Reinforcement Learning Using Equivariance

## Abstract

Model-based reinforcement learning (MBRL) is a promising approach for learning effective policies in a data-efficient manner by using learned dynamics models to generate synthetic rollouts for actor-critic trianing, thereby reducing the reliance on costly environment interactions. However, when the learned dynamics model is inaccurate, these synthetic rollouts can introduce bias and deteriorate performance. Fortunately, many domains exhibit symmetries that can serve as powerful inductive biases, enabling the learned models to generalize beyond their training data. In this work, we exploit these inherent symmetries in MBRL and formally define equivariant MBRL for POMDPs. Building on this formulation, we introduce EQUIDREAMER, a framework that integrates symmetry into both world modeling and policy learning through an equivariant latent dynamics architecture. Experiments on visual continuous control tasks demonstrate that our equivariant MBRL method outperforms both model-based and model-free baselines, achieving strong results with substantially fewer environment interactions.

## 1 Introduction

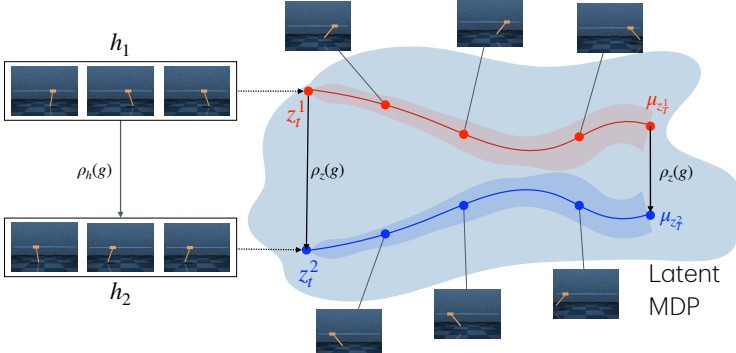

Figure 1: The POMDP has symmetries expressed by group G transformations with group actions $\rho_h$ on the history. We learn a latent MDP, where the latent states $z^i$ reflect the symmetries of the POMDP and are transformed with group action $\rho_z(g)$. Given a sequence of actions $a_{t:T}$, we can roll out trajectories that are symmetric in distribution, given symmetric starting latent states.

The prohibitive data requirements of reinforcement learning (RL) algorithms render RL impractical in many continuous control tasks, especially those involving partial observability, where the agent lacks access to the environment's underlying state and can only observe partial information, such as images (Dulac-Arnold et al., 2019). To address this limitation, MBRL methods (Janner et al., 2019; Chua et al., 2018; Hansen et al., 2022) present an attractive approach to solve the sample complexity issue by learning a predictive model of the environment's dynamics from real-world data and use it to either plan future actions (Hansen et al., 2022; Chua et al., 2018) or generate synthetic experiences for actor-critic training (Janner et al., 2019; Hafner et al., 2019a). Despite these advances, MBRL methods still require orders of magnitude more data than what is feasible in many real-world applications. In particular, learning a transition model from scratch using image inputs

remains a challenge, often resulting in lengthy training processes while being limited to addressing short-horizon tasks (Wu et al., 2023). Furthermore, world models seem to be unable to learn physical laws regardless of the amount of data used to train them (Motamed et al., 2025; Kang et al., 2024), which raises the question of how to encode such knowledge of physics into our learned model.

One promising direction is to leverage environment symmetries as inductive biases within model-based RL. As shown in Figure 1, the symmetry of the partially observable world model is evident; the learned latent dynamics model should be designed to reflect this symmetry. By constraining the hypothesis space to respect known properties of the environment, we can guide learning toward more plausible models, helping the model learn with less data. Furthermore, by encoding the symmetry in the model's architecture, we ensure that the model inherently exhibits the desired behavior of reflecting the environment's underlying symmetry. The incorporation of environment symmetries in RL algorithms has demonstrated strong performance in model-free RL (Wang et al., 2022a; Zhu et al., 2023), and imitation learning scenarios (Wang et al., 2024; Huang et al., 2022; 2024a;b). Furthermore, recent papers have shown the benefits of equivariant representations within world models for fully observable MDP settings (Park et al., 2022; Zhao et al., 2024). However, none of these methods exploit the symmetries of model-based RL under partial observability (POMDPs).

In this paper, we present EQUIDREAMER, a model-based reinforcement learning framework that exploits equivariant architectures to capture inherent environmental symmetries. Specifically, our method leverages a pretrained encoder and frame averaging to efficiently learn equivariant latent states over history. Leveraging these representations allows us to realize equivariant policies that achieve superior sample efficiency and generalization. Rather than using image reconstruction as in Dreamer (Hafner et al., 2025), which is often expensive and inefficient, our method reconstructs in the equivariant feature space, resulting in more efficient computation and improved robustness in model-based RL. Our contribution can be summarized as: 1). We analyze the symmetries underlying model-based RL and propose equivariant MBRL for POMDPs; 2). We introduce EQUIDREAMER, which leverages these symmetries through equivariant architectures and improves reconstruction by operating in the equivariant feature space; 3). We demonstrate the effectiveness of our method on visual continuous control tasks, comparing it against both model-based and model-free RL algorithms on the DeepMind Control benchmark.

## 2 PRELIMINARIES

**Group-Invariant POMDP** A partially observable Markov decision process (POMDP) is defined by the tuple $(\mathcal{S}, \mathcal{A}, \mathcal{O}, O, T, R, H, \gamma)$, where $\mathcal{S}$, $\mathcal{A}$, and $\mathcal{O}$ are the state space, action space, and observation space, respectively. $O : \mathcal{S} \times \mathcal{A} \to \mathcal{O}$ is the emission function, and $T : \mathcal{S} \times \mathcal{A} \to \mathcal{S}$ is the transition function. The complexity in acting in POMDPs stems from the fact that an optimal agent needs to choose its actions $\pi(.|h_t)$ based on the entire history $h_t = (o_0, a_0, \ldots, a_{t-1}, o_t)$ (Kaelbling et al., 1998). Denoting the space of all histories as $\mathcal{H}$, the goal is to find a history-policy $\pi : \mathcal{H} \to \mathcal{A}$ which maximizes the expected discounted return

$$J = \mathbb{E}\left[\sum_{t=0}^{\infty} \gamma^t R(s_t, a_t)\right],$$

where $\gamma \in [0, 1)$ is a discounting factor. Nguyen et al. (2023) introduced the formulation of group-invariant POMDPs. A POMDP is said to be group-invariant under a group $G$ if its transition, reward, and emission functions are invariant under the group action. Specifically, for all $g \in G$, the following conditions must hold: $T(gs, ga, gs') = T(s, a, s')$, and $R(gs, ga) = R(s, a)$.

**Statement 1** *By extending the group action to the entire observation–action history, $gh_t := \{go_0, ga_0, \ldots, ga_{t-1}, go_t\}$, Nguyen et al. (2023) shows that the optimal value function is* invariant *to group transformations, i.e., $V^*(gh) = V^*(h)$, and the optimal policy is* equivariant*, i.e., $\pi^*(gh) = g\pi^*(h)$.*

**Dreamer.** Hafner et al. (2019b) presented the recurrent state space model (RSSM) to model POMDPs and use them in an MPC scheme. Dreamer (Hafner et al., 2019a) used the RSSM as a learned simulator to learn a policy $\pi_\theta$ and a value function $v_\psi$. The RSSM is a latent dynamics model that learns a latent state representation $z_t$, a transition function $p_\theta(z_{t+1}|z_t, a_t)$, an observation model $p_\theta(o_t|z_t)$, and a reward function $p_\theta(r_t|z_t)$. The model uses a modified version of the

evidence lower bound objective (ELBO) to learn a forward model suitable for control and consists of

- RNN $h_t = f_\xi(h_{t-1}, z_{t-1}, a_{t-1})$
- Inference Network $z_t \sim q_\phi(z_t|h_t, o_t)$
- Transition Network $\hat{z}_t \sim p_\theta(\hat{z}_t|h_t)$
- Decoder $\hat{o}_t \sim p_\theta(o_t|h_t, z_t)$.

The RSSM represents latent states with a deterministic RNN state $h_t$ and a stochastic variable $z_t$, and is trained by maximizing the objective

$$\mathcal{L}(\phi, \theta) \doteq \mathbb{E}_{q_\phi}\left[\sum_{t=1}^{T}(\beta_{\text{pred}}\mathcal{L}_{\text{pred}} + \beta_{\text{dyn}}\mathcal{L}_{\text{dyn}} + \beta_{\text{rep}}\mathcal{L}_{\text{rep}})\right],$$

$$\text{where, } \mathcal{L}_{\text{pred}} = -\ln p_\theta(o_t \mid z_t, h_t) - \ln p_\theta(r_t \mid z_t, h_t)$$

$$\mathcal{L}_{\text{dyn}} = \text{KL}\left[\text{sg}(q_\phi(z_t \mid h_t, x_t)) \parallel p_\theta(z_t \mid h_t)\right], \ \mathcal{L}_{\text{rep}} = \text{KL}\left[q_\phi(z_t \mid h_t, x_t) \parallel \text{sg}(p_\theta(z_t \mid h_t))\right].$$

The objective function consists of three terms: $\mathcal{L}_{\text{pred}}$ is the reconstruction loss, $\mathcal{L}_{\text{dyn}}$ optimizes the transition model applying the stop gradient operator (sg) to $q_\phi$ and $\mathcal{L}_{\text{rep}}$ regularizes the inference network to not be over-reliant on the observations reconstruction applying sg to $p_\theta$. Dreamer uses the equivariant latent dynamics model to generate model rollouts to train an equivariant policy and an invariant value function, where the value function is trained using TD($\lambda$) returns as targets

$$\min_\phi \mathbb{E}_{z_{t'} \sim p_\psi, a_{t'} \sim \pi_\theta}\left[\sum_{t'=t}^{t+H}\frac{1}{2}\big\|v_\phi(z_{t'}) - R(z_{t'})\big\|^2\right], \text{ where}$$

$$R(z_t) = r_t + \gamma(1 - d_t)\left((1 - \lambda)v_\phi(z_t) + \lambda R(z_{t+1})\right), \ R_{t+H} = v_\phi(z_{t+H}),$$ 

(1)

and $H$ is the horizon of the model trajectories. The policy $\pi$ is trained with an objective using TD($\lambda$) returns

$$\max_{\pi_\theta} \ \mathbb{E}_{a_{t'} \sim \pi, z_{t'} \sim p_\theta}\left[R(z_t) \mid z_t\right].$$ 

(2)

**Equivariant Networks and Equivariance Learning** In the context of equivariant neural networks, a group $G$ is a mathematical structure that captures symmetries in data (Bronstein et al., 2021). A group representation maps a group $G$ to some d-dimensional general linear (GL) group $\rho : G \rightarrow GL_d$ where each element $g \in G$ is mapped to a representation $\rho(g) \in \mathbb{R}^{d \times d}$, in this context, $\rho$ how the group $G$ acts on the inputs. Given a function $f : X \rightarrow Y$, we have the representations $\rho_X$ and $\rho_Y$ acting on $X$ and $Y$, we say that a function is equivariant if $\forall x \in X, \ g \in G, \ f(\rho_X(g) \cdot x) = \rho_Y(g) \cdot f(x)$ and say that the function is invariant if, for all $x \in X, \ g \in G, \ f(\rho_X(g) \cdot x) = \cdot f(x)$. In this work, we assume that $\rho_X$ and $\rho_Y$ are known. However, this is not always the case in real-world applications (Park et al., 2022). eIn this paper, we investigate planar symmetries; therefore, we are primarily concerned with actions from the $D_n$ and $C_n$ groups, where $gx$ the group action on a the signal $x$ is a rotation or flipping in the spatial dimension $\rho_{spatial}(g)^{-1}x$ followed by a transformation in the channel dimension $g(x) =$

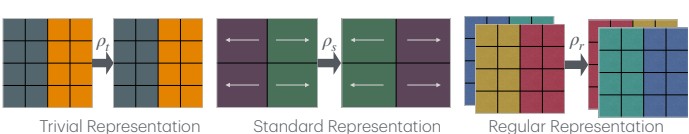

Trivial Representation     Standard Representation     Regular Representation

Figure 2: Illustration of the different group actions considered in our work using the horizontal flip group action as described in section 2.

$\rho(g)(\rho_{spatial}(g)^{-1}x)$, where $\rho(g)$ depends on the representation of the signal. Our method considers 3 types of representations, which we illustrate in figure 2. **Trivial representation actions** $\rho_t$, this representation applies the identity transformation to the channel dimension. For example, when rotating an image, the pixel values remain unaffected by the transformation. **Standard representation actions** $\rho_s$ is usually used with data representing Cartesian vectors; **Regular representation actions** $\rho_r$ get applied through permutation in the channel dimension as highlighted in figure 2.

## 3 EQUIVARIANT MODEL-BASED REINFORCEMENT LEARNING

This work primarily focuses on the POMDP formulation. Building on Statement 1 of Group-Invariant POMDPs, our goal is to learn an equivariant optimal policy within a model-based RL framework. Specifically, we employ an equivariant latent dynamics model to capture the invariant POMDP structure, and generate rollouts from this model to train an equivariant policy. In this section, we first show that the optimal transition function in model-based RL in invariant POMDPs is group-invariant, and then describe our approach to realizing these equivariant properties.

The performance of MBRL methods depends on their ability to accurately estimate the value of a given policy $\pi$. The simulation lemma (Kolev et al., 2024; Abbeel & Ng, 2005; Strehl & Littman, 2008) bounds the difference the policy evaluation in the model and the policy evaluation in the true dynamics given. We use the formulation from Kolev et al. (2024)

$$|J_{p^*}(\pi) - J_{p_\theta}(\pi)| \leq \frac{\gamma R_{\max}}{(1-\gamma)^2} \mathbb{E}_{(s,a) \sim d^\pi_{p_\theta}} \left[ \mathbb{D}_{TV}(p_\theta(\cdot \mid s, a), p^*(\cdot \mid s, a)) \right], \tag{3}$$

formalizes this connection by bounding the error in value estimation $|J_{p^*}(\pi) - J_{p_\theta}(\pi)|$ in terms of the distance between the learned dynamics model $p_\theta$ and the true environment dynamics $p^*$.

**Proposition 1** *An invariant dynamics model $p_\theta(gz_t|gz_{t-1}, ga_{t-1}) = p_\theta(z_t|z_{t-1}, a_{t-1})$, that minimizes the training error on the visited states and incorporates the underlying symmetry of the ground-truth POMDP leads to lower distance $|J_{p^*}(\pi) - J_{p_\theta}(\pi)|$ in comparison to an unstructured learned dynamics model.*

Intuitively, in an equivariant POMDP, the ground-truth dynamics are invariant, as discussed in Section 2. Introducing structure and learning an invariant dynamics model would thus effectively prune the hypothesis space for optimization, and the additional generalization benefits of a symmetric transition would lead to a lower bound on policy evaluation given the true dynamics. Detailed proof could be found in Appendix A. Based on Proposition 1, we propose the use of equivariant model-based RL for solving equivariant POMDPs. The objective of equivariant MBRL is to obtain an equivariant policy $\pi_\theta(z_t)$ together with an invariant value function $v_\theta(z_t)$ as discussed in section 2. Incorporating symmetries into the learned solution reduces model error on the training distribution and enhances generalization to equivalent states that are unseen during training.

## 4 EQUIDREAMER

In this section, we present our method for realizing equivariant model-based RL, namely EQUIDREAMER. It consists of an equivariant RSSM for learning the world model and an equivariant model–based actor–critic using data generated from the equivariant RSSM.

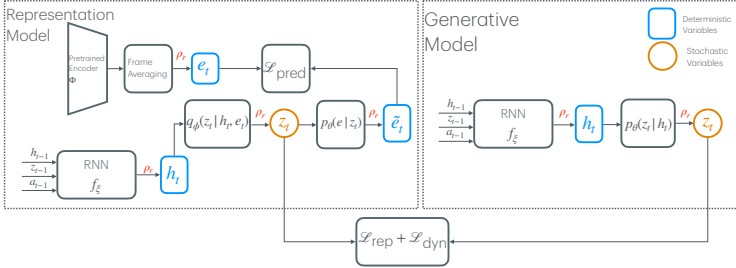

Figure 3: Architecture of equivariant RSSM, $\rho_r$ is the regular representation, and $\rho_t$ is the trivial representation. The pretrained encoder delivers encoding in the regular representation after frame averaging which are then used both as input to the inference network and to provide reconstruction targets to learn equivariant representations.

## 4.1 EQUIVARIANT RECURRENT STATE SPACE MODEL

We consider a POMDP with underlying states $s_t$ that exhibit symmetries in group $G$. To capture these, we introduce a latent MDP with Markovian states $z_t$. We view $z_t$ as feature vector fields on which the group action $\rho_z(g)$ operates, enabling the encoding of environmental symmetries, as illustrated in figure 1 and related to MDP homomorphism (Van der Pol et al., 2020). To realize this latent formulation, we design a sequential latent variable model (Girin et al., 2020) and realize the properties of group-invariant POMDPs from Section 2, where both the reward and transition functions remain invariant under group actions. Specifically, we adopt the Recurrent State-Space Model (RSSM) (Hafner et al., 2019b). The architecture of our equivariant variant is shown in Figure 3. We next describe the representation and generative models that compose our latent dynamics learning process.

### 4.1.1 REPRESENTATION MODEL

The representation learning process aims to learn a state representation that contains sufficient information to reconstruct both observations and rewards. Given a history sequence $\{o_1, a_1, r_1, \ldots o_H, a_H, r_H\}$, we learn an image encoder $e_t = \Psi(o_t)$, a recurrent neural network $f_\xi(h_{t-1}, z_{t-1}, a_{t-1})$, an inference model $q(z_t|h_t, e_t)$, an embedding decoder $p(e_t|z_t, h_t)$ and a reward function $p(r_t|z_t, h_t)$. Each component is designed to reflect the properties of the invariant POMDP as introduced in Section 2.

**Image Encoder** Assume the image observations are transformed; the corresponding latent features should change accordingly. Thus, we aim to learn an image encoder satisfying

$$\Psi(\rho_t(g)o_t) = \rho_r(g)\Psi(o_t). \tag{4}$$

To realize the equivariance of $\Psi$, we adopt the symmetrization technique (Puny et al., 2021), using a non-equivariant function $\psi(o_t)$ and averaging its outputs over group-transformed versions of the image observations.

$$\Psi(o_t) = \frac{1}{|G|} \sum_{g \in G} \rho_y(g)\psi(\rho_t(g)^{-1}o_t). \tag{5}$$

This allows us to leverage pretrained encoders and achieve strong representations, as demonstrated by Wang et al. (2025).

**Posterior Transition** The transition in the RSSM relies on a deterministic hidden state $h_t$ from the RNN, and a stochastic latent state $z_t$. Both states need to be represented in the regular representation, $\rho_r$, to provide an equivariant input to the decoders, the policy, and the critic. Thus, the invariant transition model is implemented to be equivariant. For an equivariant $h_t$, we leverage the architecture of the equivariant LSTM presented in (Nguyen et al., 2023) to model the equivariant RNN

$$f_\xi(\rho_r(g)h_{t-1}, \rho_r(g)z_{t-1}, \rho_a(g)a_{t-1}) = \rho_r(g)h_t.$$

We describe the architecture for the equivariant RNN in detail in Appendix C. The equivariant inference network $q_\phi$ models the stochastic latent variable $z_t$ as a heteroscedastic Gaussian distribution $\mathcal{N}(\mu_{z_t}; \Sigma_{z_t})$, where we model both the mean $\mu_{z_t}$ and the diagonal variance $\Sigma_{z_t}$ in the regular representation $\rho_r$, thus satisfying

$$q_\phi(z_t|\rho_r(g)h_t, \rho_r(g)e_t) = \mathcal{N}(\rho_r(g)\mu_{z_t}; \rho_r(g)\Sigma_{z_t}). \tag{6}$$

We observe in our experiments that Gaussian latent variables are more amenable to equivariance compared to the categorical latent variables used by default in DreamerV3. We hypothesize that sampling from a latent state distribution is inherently not equivariant, as the random noise used in the reparameterization trick is independent of the input. Thus, sampling from the equivariant latent state distribution is not equivariant. However, samples from the unimodal Gaussian distribution are likely to be similar to the mean and benefit more from equivariance compared to the discrete multimodal categorical distribution.

**Decoder** In the RSSM, the model is supervised using observation reconstruction, which was shown to be more performant for the specific case of the RSSM in (Hafner et al., 2019a). Further, (Ni et al., 2024) empirically showed the benefit of using observation reconstruction as an auxiliary loss in partially observable settings, albeit in a model-free setting. In EQUIDREAMER, learning equivariant latent representations would necessitate the observation reconstruction to also be equivariant. In the case of images, this would require maintaining a spatial dimension in both the deterministic state $h_t$ and the stochastic state $z_t$, thereby incurring memory-intensive operations. To avoid the complications of image reconstruction and maintain an equivariant architecture, we follow a similar approach to Assran et al. (2023) and reconstruct learned embeddings using an embedding predictor $e_t = f(z_t, h_t)$, which satisfies the equivariance property as the target embeddings are made equivariant using frame averaging $ge_t = f(gz_t, gh_t)$.

Reconstructing embeddings instead of images has two advantages: the first is predicting a low-dimensional vector instead of the full image, which is more computationally efficient. In addition, reconstructing feature embeddings is less prone to being distracted by various possible noise in the image (Assran et al., 2023). A natural extension of this would be to predict the features from the image encoder $f_\psi$; however, as the encoder is updated during training, we would need to resort to stabilizing techniques, such as a target network, as done in (Grill et al., 2020). Instead, we resort to predicting the features of a pretrained network as done in (Zhou et al., 2024; Zhai et al., 2025).

### 4.1.2 GENERATIVE MODEL

The generative model is concerned with learning a transition network $p_\theta(z_t|h_{t-1}, z_{t-1}, a_{t-1})$ which generates latent states depending on the previous latent state and policy actions. This synthetic data is then used in actor-critic training.

**Prior Transition** The transition model $p_\theta$ is implemented as an equivariant MLP that takes as input previous latent states in the regular representation and outputs a new latent state also in the regular representation. In the generative model we use the same RNN used in representation learning and implement $p_\theta$ analogous to the posterior transition learning in equation 6.

### 4.1.3 TRAINING

For training the equivariant RSSM, we optimize a prediction loss $\mathcal{L}_{\text{pred}}$ as well as the KL terms $\mathcal{L}_{\text{dyn}}$ and $\mathcal{L}_{\text{rep}}$. We use the same KL terms as the RSSM and adapt $\mathcal{L}_{\text{pred}}$ to embedding prediction. We implement the prediction loss as

$$\mathcal{L}_{\text{pred}} = (1 - \cos(\hat{e}_t, e_t)) - \log p(r_t|z_t, h_t), \tag{7}$$

where we use the cosine distance between the reconstructed embedding $\hat{e}_t$ and the frame averaging encoding $e_t$. We find that the cosine distance leads to more stable optimization compared to the L2 distance. We also explored following the approach in DreamerPro (Deng et al., 2022) and investigated an equivariant version of SwAV (Caron et al., 2020). We find that a feature reconstruction has lower implementation complexity and mitigates the image augmentation process required by the DreamerPro model. We compare the equivariant versions of both methods in our ablation studies in Section 5.

### 4.2 EQUIVARIANT MODEL-BASED ACTOR-CRITIC

We train the policy $\pi_\theta$ and value function $v_\phi$ using model-generated rollouts, as described in Section 2. The policy is modeled as a stochastic Gaussian policy designed to be equivariant, following the approach of Wang et al. (2022c). Specifically, we enforce equivariance in the policy mean such that $\pi_\theta(gz_t) = \mathcal{N}(\rho_a(g)\mu; \rho_t(g)\Sigma)$., where $\rho_a(g)$ denotes a group action for the task's action space and $z_t$ is the latent state. The implementation of equivariance depends on environment-specific details. The policy's standard deviation is modeled in the trivial representation $\rho_t$, ensuring it remains invariant to group actions. We optimize the policy by maximizing the objective in Equation 2. The value function is designed to be invariant under group transformations, i.e., $v_\phi(gz_t) = v_\phi(z_t)$, aligning with the group invariance property of the underlying POMDP. It is trained following the approach used in Dreamer, by minimizing the objective in Equation 1 using model rollouts. Further implementation details are provided in Appendix D.

## 5 RESULTS

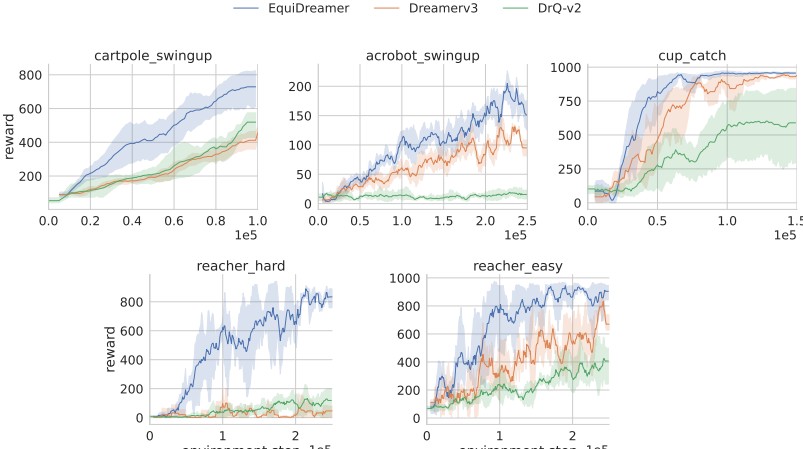

Figure 4: Results of our method EQUIDREAMER compared to DreamerV3, DrQ-v2. We observe that our method learns faster than Dreamer and DrQ-v2 in all environments, especially in the more challenging reacher-hard environment, primarily due to replacing image reconstruction with feature reconstruction and the additional inductive biases encoded in the model architecture, despite the background in the DMC environments corrupting the symmetry in the observations.

In our experiments, we compare our approach EQUIDREAMER with a model-based baseline in **Dreamerv3** and a model-free baseline **DrQ-v2**. Both methods provide very strong RL baselines for RL with visual observations. We evaluate our methods and the baselines in DMC environments with symmetry. We evaluate our method on the CartPole, Acrobot, and Cup Catch environments, all of which are symmetric with respect to vertical-axis flipping. Additionally, we consider the Reacher environment, which is symmetric on both the vertical and horizontal axes. To evaluate the performance of EQUIDREAMER on robot learning tasks, we compare our method to DreamerV3 on fully observable robot learning tasks from Wang et al. (2022b). Further, we evaluate on a partially observable robot learning task from Nguyen et al. (2023). In the partially observable task, the agent must actively gather information on whether the object is movable and retain in its memory its past interactions with the objects to determine the next action. As previously discussed in Wang et al. (2022d); Nguyen et al. (2023). The tasks' dynamics are invariant to changes in the reference frame, thereby inducing SO(2) equivariance.

### 5.1 DMC SUITE EXPERIMENTS

The results in Figure 4 show that EQUIDREAMER, using equivariance and replacing the image reconstruction loss, significantly improves the performance of Dreamerv3. This effect is more pronounced in the reacher-hard environment, which is the most challenging task in our environments. These results demonstrate empirically that encoding symmetry in the architecture enables faster learning of representations, improving sample efficiency. Furthermore, avoiding the reconstruction of raw observations can improve performance in more challenging environments, as highlighted in the results in the reacher environments. We run each algorithm for five seeds, and evaluate the episodic returns for five evaluation episodes each. The curves represent a rolling window of ten evaluations, and we also display the 95% confidence interval. We ran all of our experiments on A100s.

### 5.2 ROBOT MANIPULATION EXPERIMENTS

We consider the robot manipulation tasks from the BulletArm benchmark Wang et al. (2022b). The agent has a top-down depth camera observation, as shown in 6. In our manipulation experiments, the implementation of the equivariant policy differs as each element in the action vector transforms differently as discussed in Wang et al. (2022d); Nguyen et al. (2023). The action vector consists

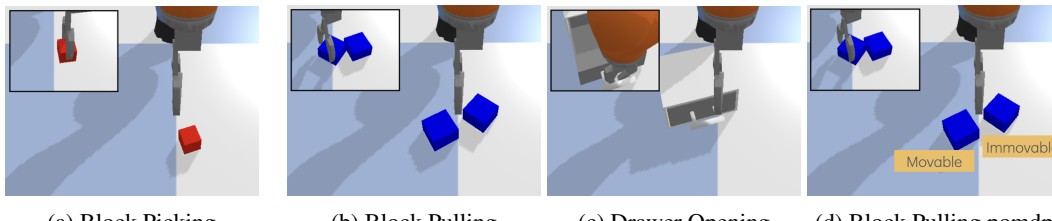

| (a) Block Picking | (b) Block Pulling | (c) Drawer Opening | (d) Block Pulling pomdp |

Figure 5: The robot manipulation environments investigated in our work. The block pulling POMDP environment differs from the fully observable version in that one of the blocks is immovable, and the agent can only get this information by pulling the block, thus the agent needs to explore to gather information and retain information from its previous observations.

of $(w, x, y, z, r)$, where $(x, y, z)$ are the cartesian coordinates and $r$ is the gripper opening command and $w$ is the gripper angle are represneted in $\rho_t$ as they do not change by $SO(2)$ rotation, and $(x, y)$ are both represented using the standard representation $\rho_s$. The results in Figure 8 show that EquiDreamer also leads to improvements in robot manipulation environments with $SO(2)$ symmetry, even though we are implementing equivariance in our model using the C4 group and with no augmentation, as usually done in equivariant RL literature. Due to the use of the RSSM as a base model, which relies on a recurrent neural network in its transition function, we see that our method can solve the partially observable block pulling task with better efficiency than Dreamerv3.

## 5.3 ABLATION STUDIES

### 5.3.1 CHOICE OF SUPERVISION FOR EQUIVARIANT DYNAMICS MODEL

In figure 7, we compare our method in supervising the equivariant RSSM by reconstructing the observation embeddings in the regular representation by using a pretrained encoder and frame averaging with the approach followed in DreamerPro (Deng et al., 2022), where the corresponding observations and the latent states are clustered together instead of reconstructing the whole image. We observe in Figure 7 that embedding reconstruction yields significantly faster learning in the more complex reacher environment; however, in a relatively easy environment like CartPole, the difference is less pronounced.

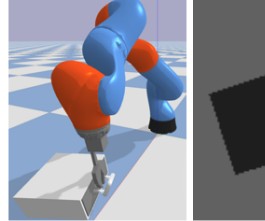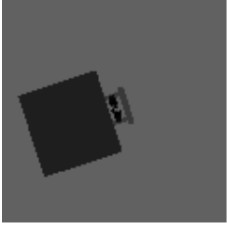

Figure 6: The robot arm and its depth image from the drawer opening environment Wang et al. (2022b). The agent observes a top-down depth image, making the environment $SO(2)$ equivariant from the standpoint of the robot.

### 5.3.2 EFFECT OF REPLACING IMAGE RECONSTRUCTION

To disentangle the effect of the equivariant architecture and the effect of replacing the image reconstruction objective. We replace the encoder in the Dreamerv3 model with the resnet-18 pretrained network used in our equivariant model and replace the image reconstruction loss with an encoding reconstruction loss. This architecture is similar to our model except for equivariance. The results in figure 9 illustrate that equivariance is the reason for improvement in the three environments with D1 group symmetry. However, in the reacher environment, it seems that most of the performance improvement is down to the replacement of the image reconstruction objective with embedding reconstruction. Generally, it appears that reconstructing images is counterproductive in more challenging tasks. It should be noted that, in our approach, we reconstruct in the feature space of a frozen pretrained ResNet-18, which might not necessarily capture all the task-relevant details, which could explain the performance inconsistency compared with vanilla Dreamer. However, we find in Table 1 that replacing image reconstruction with a smaller model; this effect would be even more pronounced in more realistic situations where we need to reconstruct higher-dimensional images than those used in our experiments.

## 6 RELATED WORK

**Model-based RL.** Model-based reinforcement learning (MBRL) leverages a learned forward dynamics model to improve sample efficiency in decision-making, building on theoretical foundations such as the simulation lemma. Early work like PILCO (Deisenroth & Rasmussen, 2011) used Gaussian processes to model uncertainty and propagate it during planning. Extending this idea, PETS (Chua et al., 2018) employed ensembles of neural networks to achieve more scalable, uncertainty-aware planning. MBPO (Janner et al., 2019) combined learned models with off-policy actor-critic methods to enhance sample efficiency. While these methods typically assumed access to

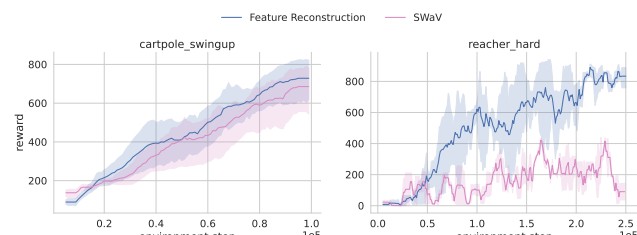

Figure 7: Comparing our approach in supervising the equivariant model in reconstructing equivariant embeddings with the approach followed in DreamerPro (Deng et al., 2022) by using SWaV (Caron et al., 2020). Using SWaV is sufficient in a simple environment, like CartPole. However, it struggles in more challenging environments, like Reacher.

low-dimensional state spaces, latent variable models extended MBRL to high-dimensional observations such as images. For example, Embed-to-Control (E2C) (Watter et al., 2015) and DVBF (Karl et al., 2016) used linear latent dynamics, while Dreamer (Hafner et al., 2019a) introduced recurrent state-space models (RSSMs) capable of handling both high-dimensional inputs and partial observability (Pasukonis et al., 2022). These were trained jointly with on-policy actor-critics and have demonstrated strong performance across various environments, including the challenging Minecraft benchmark (Hafner et al., 2025). TD-MPC (Hansen et al., 2022) further combined latent-variable models with actor-critic training and model-predictive control (MPC), achieving strong performance in both simulation and real-world robotic settings. Both Dreamer and TD-MPC variants have been validated on real robotic platforms (Wu et al., 2023; Lancaster et al., 2024), outperforming model-free counterparts in sample efficiency and task success.

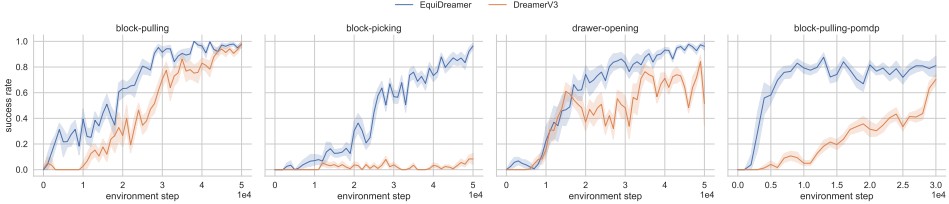

Figure 8: Results of EquiDreamer compared to DreamerV3. In the robot manipulation environments described in section 5, we observe that our method improves Dreamer's performance on fully observable tasks and on partially observable block-pulling, where the agent must discover which block is movable before moving the blocks.

**RL in POMDPs.** A significant portion of reinforcement learning research in POMDPs has focused on incorporating recurrence into standard actor-critic frameworks. For instance, Recurrent DQN (Hausknecht & Stone, 2015) augments Deep Q-Networks with recurrent neural networks to maintain an implicit belief over hidden states, enabling more effective decision-making under partial observability. Later work (Ni et al., 2021) demonstrated the superiority of recurrent methods across a wide range of POMDP benchmarks, outperforming baselines specifically designed for such settings. These findings support the use of recurrence as a general-purpose mechanism for handling partial observability, which may help explain Dreamer's strong performance in POMDP environments, where the transition function is partially implemented as an RNN (Pasukonis et al., 2022). More recently, transformer-based architectures have achieved state-of-the-art results in POMDPs, surpassing recurrent models due to the increased capacity of self-attention compared to a single hidden vector used in RNNs. This shift has motivated architectures that combine structured sequence

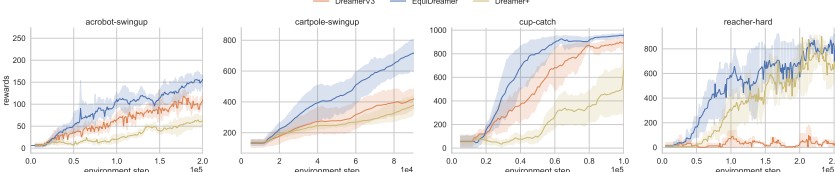

Figure 9: To single out the effect of equivariance, we replace the image reconstruction objective in Dreamerv3 with an embedding reconstruction objective and replace the encoder with a pretrained ResNet-18 encoder. We name the modified architecture **Dreamer⁺**. Although EquiDreamer outperforms Dreamer⁺ in most environments, the difference is less pronounced in the reacher-hard environment, thus indicating that the performance gains in that environment are mostly down to the replacement of the image reconstruction objective.

models such as transformers or state-space models (SSMs) with the Dreamer framework, further improving performance under partial observability (Deng et al., 2023; Samsami et al., 2024). These architectural advances are largely complementary to our approach. In this work, rather than focusing on architectural innovations for sequence modeling, we explore integrating inductive biases into the latent variable model.

**Equivariance in RL.**   Recent work explored the integration of equivariant architectures within reinforcement learning frameworks. For instance, (Wang et al., 2022d) incorporated equivariant representations into traditional actor-critic algorithms, showing better generalization and sample efficiency. Equivariance has also been successfully applied in on-robot learning (Wang et al., 2022a; Zhu et al., 2023). More recently, Wang et al. (2024) combined equivariant architectures with diffusion models for learning from offline data. In the model-based setting, Zhao et al. (2024) introduced an equivariant world model, but their method is limited to fully observed environments and employs random-sampling-based planning strategies that are difficult to align with equivariant structure. EDGI Brehmer et al. (2023) is an offline method that uses diffusion to learn a dynamics model and a planner building on diffuser Janner et al. (2022); however, the method requires an excessive number of demonstrations and does not extend to POMDPs. EQR Mondal et al. (2022) is a model-free method in which learning the transition model is used only to supervise the learning of equivariant representations; it thus doesn't use model rollouts for learning, but rather to study the problem of learning equivariant representations that are unknown a priori. SEN Park et al. (2022) studies the case of learning representation learning for world models, where the symmetry in the observation space might not be entirely known, while the symmetry in the latent space is known. We consider the method to be orthogonal to our work. Our work builds on these foundations and extends the equivariant recurrent architecture proposed in (Nguyen et al., 2023) to the model-based RL setting. We propose an approach combining world models that can handle partial observability and group equivariance in a unified framework. This enables sample-efficient learning in robotic manipulation tasks while retaining the generality and scalability required for real-world deployment.

## 7 CONCLUSION

In this paper, we present a novel algorithm that combines the benefits of equivariant architectures with the sample efficiency of MBRL. By incorporating equivariance into the architecture of a latent dynamics model, our approach enables the model to exploit the symmetries inherent in the underlying POMDP. We explore several methods for supervising the dynamics model to learn equivariant latent state representations and demonstrate the effectiveness of our approach on DMC visual control tasks. While our experiments focus on environments with planar symmetries, many real-world tasks involve more complex, three-dimensional symmetries. Extending our architecture to capture such symmetries is a promising direction for future research. Additionally, we investigate continuous latent state distributions in our model, and extending this work to support sparse latent state distributions presents another interesting research direction.

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

# A  PROOF OF PROPOSITION 1

We prove that an equivariant dynamics model $p_\theta^{\text{equi}}$ yields a smaller policy evaluation error $|J_{p^*}(\pi) - J_{p_\theta}(\pi)|$ compared to an unstructured learned dynamics model $p_\theta^{\text{non-equi}}$.

We begin with the simulation lemma, using $x = (s, a)$ for notational simplicity:

$$|J_{p^*}(\pi) - J_{p_\theta}(\pi)| \leq \frac{\gamma R_{\max}}{(1-\gamma)^2} \mathbb{E}_{(x) \sim d_{p_\theta}^\pi} \left[ \mathbb{D}_{TV}(p_\theta(\cdot \mid x), p^*(\cdot \mid x)) \right]$$

Note that we are considering $d_{p_\theta}^\pi$, since we generate data for the actor-critic training via rollouts in $p_\theta$ using real-world samples $p(s_0) \sim d_{p^*}^\pi$ as starting states. We partition the state-action distribution $d_{p_\theta}^\pi(x)$ into three disjoint components:

- $d_v^\pi(x)$: distribution over states visited during training $p_\theta$.
- $d_s^\pi(x)$: distribution over unvisited states that are symmetric to in distribution samples to $p_\theta$.
- $d_u^\pi(x)$: distribution over unvisited states with no symmetric samples in $p_\theta$'s straining data.

Although we roll out in the learned model $p_\theta$, the agent might still find itself in $d_s^\pi(x)$ and $d_u^\pi(x)$ due to the fact that the starting state might lie in either region of the state space. This decomposition allows us to rewrite the bound as:

$$\begin{aligned}
|J_{p^*}(\pi) - J_{p_\theta}(\pi)| \leq \frac{\gamma R_{\max}}{(1-\gamma)^2} \Big( &\mathbb{E}_{x \sim d_{\theta,v}^\pi} \left[ \mathbb{D}_{TV}(p_\theta(\cdot \mid x), p^*(\cdot \mid x)) \right] \\
&+ \mathbb{E}_{x \sim d_{\theta,s}^\pi} \left[ \mathbb{D}_{TV}(p_\theta(\cdot \mid x), p^*(\cdot \mid x)) \right] \\
&+ \mathbb{E}_{x \sim d_{\theta,u}^\pi} \left[ \mathbb{D}_{TV}(p_\theta(\cdot \mid x), p^*(\cdot \mid x)) \right] \Big)
\end{aligned}$$

Given we use empirical risk minimization to find the hypothesis $\hat{\theta}$, there exists a generalization gap between the test objective $L(\hat{\theta})$ evaluated on the test samples and the objective $\hat{L}(\hat{\theta})$ evaluated on the training samples. Where $L(\theta) - \hat{L}(\hat{\theta}) \geq 0$. We make two assumptions

**Assumption 1** *Bounded* $\mathbb{D}_{TV}$ *on visited states: Both $p_\theta^{equi}$ and $p_\theta^{non\text{-}equi}$ have bounded $\mathbb{D}_{TV}$ on the training distribution*

$$\mathbb{E}_{x \sim d_{\theta,v}^\pi} \left[ \mathbb{D}_{TV}(p_\theta(\cdot \mid x), p^*(\cdot \mid x)) \right] \leq \epsilon_u \leq \mathbb{E}_{x \sim d_{\theta,u}^\pi} \left[ \mathbb{D}_{TV}(p_\theta(\cdot \mid x), p^*(\cdot \mid x)) \right] \quad (8)$$

**Assumption 2** *Equivariance property: The equivariant model leverages symmetry to reduce its error in symmetric unvisited states, and have bounded error in the symmetric configurations*

$$\mathbb{E}_{x \sim d_{\theta,s}^\pi} \left[ \mathbb{D}_{TV}(p_\theta^{equi}(\cdot \mid x), p^*(\cdot \mid x)) \right] \leq \epsilon_s \leq \mathbb{E}_{x \sim d_{\theta,s}^\pi} \left[ \mathbb{D}_{TV}(p_\theta^{non\text{-}equi}(\cdot \mid x), p^*(\cdot \mid x)) \right] \quad (9)$$

Assumption 1 is reasonable given the generalization gap in empirical risk minimization hypotheses Zhang et al. (2017) and the generalization ability of neural networks in the support of their training data. In assumption 2 we assume a lower error in symmetric configurations. This assumption is reasonable, as previous work on equivariance in reinforcement learning Wang et al. (2022c) shows that leveraging equivariance, even in domains where symmetry is not perfect, results in networks that better capture symmetry, because equivariance renders symmetric configurations within the data support despite imperfect observations.

**Main Result.** Under these assumptions, the policy evaluation error bounds become: For the equivariant model:

$$|J_{p^*}(\pi) - J_{p_\theta^{\text{equi}}}(\pi)| \leq \frac{\gamma R_{\max}}{(1-\gamma)^2} \left( \mathbb{E}_{x \sim d_{\theta,u}^\pi} \left[ \mathbb{D}_{TV}(p_\theta^{\text{equi}}(\cdot \mid x), p^*(\cdot \mid x)) \right] + \epsilon_u + \epsilon_s \right)$$

For the non-equivariant model:

$$|J_{p^*}(\pi) - J_{p_\theta^{\text{non-equi}}}(\pi)| \leq \frac{\gamma R_{\max}}{(1-\gamma)^2} \left( \mathbb{E}_{x \sim d_{\theta,s}^\pi} \left[ \mathbb{D}_{TV}(p_\theta^{\text{non-equi}}(\cdot \mid x), p^*(\cdot \mid x)) \right] + \right.$$

$$\left. \mathbb{E}_{x \sim d_{\theta,u}^\pi} \left[ \mathbb{D}_{TV}(p_\theta^{\text{non-equi}}(\cdot \mid x), p^*(\cdot \mid x)) \right] + \epsilon_u \right)$$

Due to the second assumption and equation 9, we conclude:

$$|J_{p^*}(\pi) - J_{p_\theta^{\text{equi}}}(\pi)| \leq |J_{p^*}(\pi) - J_{p_\theta^{\text{non-equi}}}(\pi)|. \qquad \square$$

## B  EQUIVARIANT SWAV

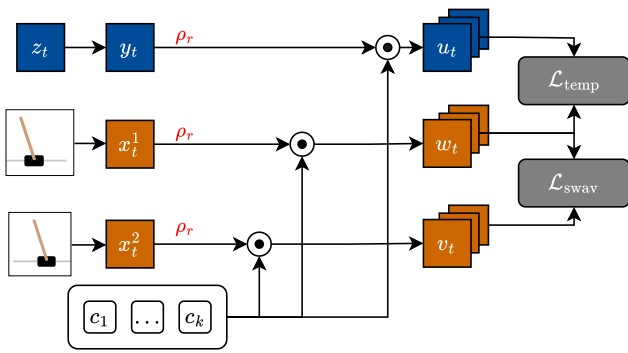

Figure 10: Architecture of equivariant SWaV. $\rho_r$ is the regular representation and $\rho_t$ is the trivial representation.

We build on the approach used in Deng et al. (2022), which extends Dreamer using the SwAV objective (Caron et al., 2020) as a proxy to the pixelwise reconstruction loss, which is shown to outperform contrastive losses and pixelwise reconstruction losses. We show our extension in figure 10. We first augment the observation $o_t^1$ to get an additional view $o_t^2$. The observation and its augmented pair $o_t^i$ are embedded into the vector $x_t^i$ in $\rho_r$, while the latent state $z_t$ is embedded into $y_t$ in $\rho_r$.

In DreamerPro, the different embeddings are clustered into one of $k$ clusters $\{c_1, \ldots, c_k\}$. In our case this would lead to group invariant clustering, i.e. if the trajectory contains two equivalent observations regarding the group transformation, an invariant clustering would encourage both states to have the same exact latent representation which would be deterimental to the policy which needs to be equivariant as discussed in section 2. Thus, we cluster each fiber channel $f$ in the regular representation of the embedding $x_t^i$ and $y_t$

$$(u_{t,1,f}^i, \ldots, u_{t,k,f}^i) = \text{softmax}\left( \frac{x_{t,f}^{(i)} \cdot c_1}{\tau}, \ldots, \frac{x_{t,f}^{(i)} \cdot c_k}{\tau} \right), \tag{10}$$

where $\tau$ is the temperature of the softmax function. Consequently, we apply the clustering via matrix multiplication with the prototypes matrix $\{c_1, \ldots, c_k\}$ on each fiber dimension separately, leading to two objectives evaluated separately for each fiber dimension

$$\mathcal{L}_{\text{swav}} = \frac{1}{2} \sum_{f=1}^F \sum_{k=1}^K \left( w_{t,k,f}^{(1)} \log v_{t,k,f}^{(2)} + w_{t,k,f}^{(2)} \log v_{t,k,f}^{(1)} \right), \tag{11}$$

$$\mathcal{L}_{\text{temp}} = \frac{1}{2} \sum_{f=1}^F \sum_{k=1}^K \left( w_{t,k,f}^{(1)} \log u_{t,k,f}^{(2)} + w_{t,k,f}^{(2)} \log u_{t,k,f}^{(1)} \right). \tag{12}$$

The embedding is clustered in clusters using matrix multiplication. We do the same thing by doing the matrix multiplication separately for each dimension in the fiber space. The objective encourages each augmented pair to belong to the same cluster. In addition, the temporal loss encourages the latent state to have the same clusters as the embeddings of the observations.

Similar to (Deng et al., 2022), $\mathcal{L}_{\text{swav}}$ helps the image embedding maintain meaningful features that are invariant to the augmentation of $o_t^2$, while $\mathcal{L}_{\text{temp}}$ encourages the latent state to maintain information that it can be assigned to the correct observation without having to reconstruct the pixel values.

## C    EQUIVARIANT RNN

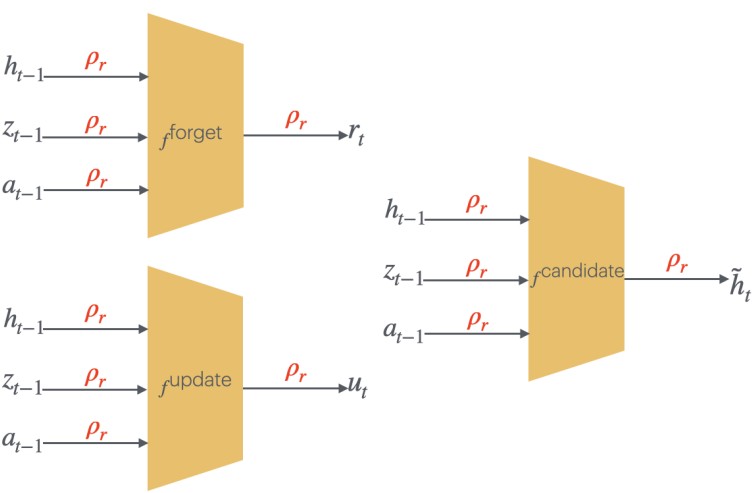

Figure 11: Architecture of equivariant gates. $\rho_r$ is the regular representation and $\rho_t$ is the trivial representation. Each gate is implemented as a $1 \times 1$ equivariant CNN and the outputs of the different gates are used to output the equivariant hidden state $h_t$ as in equation 13

The sequential modeling of the RSSM relies on a GRU cell. For our model to be equivariant, we need the GRU cells to also be equivariant such that $gf(x_{t-1}, h_{t-1}, a_t) = f(gx_{t-1}, gh_{t-1}, ga_t)$. We follow a similar architecture to (Nguyen et al., 2023), which presented an equivariant version of LSTMs. In our paper, we apply similar modifications to a GRU cell, where we replace the different gates of the RNN with equivariant versions, thus ensuring the equivariance of the whole architecture. The GRU cell consists of the following operations

$$r_t = \sigma(W_r \cdot [h_{t-1}, x_t] + b_r), \ u_t = \sigma(W_z \cdot [h_{t-1}, x_t] + b_z)$$
$$\tilde{h}_t = r_t \odot \tanh(W_h \cdot [h_{t-1}, x_t] + b_h) \tag{13}$$
$$h_t = (1 - u_t) \odot h_{t-1} + u_t \odot \tilde{h}_t \ .$$

The forget gate $r_t$ decides which parts of the hidden state are forgotten, and the update gate $z_t$ decides which parts of the hidden state need to be maintained. The gating mechanisms in LSTM and GRU cells have been effective in modeling long sequences; however, they are now superseded by the attention mechanism in transformers due to their limited expressiveness compared to attention, which is more efficient and avoids the quadratic scaling problem associated with attention. We modify the GRU cell architecture by replacing the different operations with equivariant replacements. Here, we utilize equivariant CNNs, which employ $1 \times 1$ convolutions. All the hidden states and the inputs of the different gates are in the regular representation, as shown in Figure 11. We note here that our equivariant RNN implicitly assumes fixed transformation $g$, that does not change over time, howevert this might not necessarily be the case.

# D IMPLEMENTATION DETAILS

## D.1 ADOPTION TO EQUIVARAINT ARCHITECTURE

One design choice that we took in comparison to Dreamerv3 is to rely on Gaussian latent states instead of categorical latent states. We made this decision because we only observed a performance boost with equivariance when using Gaussian latent states. We postulate that this could be the case because of the unimodality of Gaussians and the multimodality of categorical distribution, where the samples might be less likely to maintain equivariance. Further, to ensure a similar number of parameters, we scale the weights of the network, which ensures that the number of weights is similar; we list the number of weights in table 1.

| Group | model | actor | critic |
|---|---|---|---|
| Dreamerv3 | 13.1M | 1.05M | 0.92M |
| Dreamer$^+$ | 10.9M | 1.05M | 0.92M |
| EquiDreamer (D1) | 9.1M | 0.79M | 0.53M |
| EquiDreamer (D2) | 11.7M | 0.79M | 0.65M |

Table 1: Number of weights in each module for Dreamer, EquiDreamer and Dreamer$^+$

## D.2 TRAINING ALGORITHM

We use a phasic approach, where a dynamic model is first fitted to the data, and the model is then used to train the policy and value function on rollouts generated by our model. We show the training steps in algorithm 1.

---

**Algorithm 1** Pseudocode for EquiDreamer Training

---

Initialize parameters
**for** N episodes **do**
    **for** t timesteps **do**
        sample $a_t$ from policy $\pi_\theta(s_t)$
        **if** should train **then**
            Train RSSM using model data $\mathcal{D}$
            Roll out $\pi_\theta$ in forward model with horizon $H_\pi$
            Train $\pi_\theta$ and $v_\phi$ with data from model rollouts.
        **end if**
    **end for**
    Add $\mathcal{D}_{\text{episode}}$ to $\mathcal{D}$ wand
**end for**

---

## D.3 HYPERPARAMETERS

Our method involves hyperparameters for the model training and the actor-critic algorithm. The hyperparameters are mostly the same as those used in DreamerV3 (Hafner et al., 2025). For completeness, we list our hyperparameters in Table 2.

| Model | |
|---|---|
| learning rate | $1e^{-3}$ |
| number of hidden units | 512 |
| Stochastic state dimensions | 32 |
| hidden state dimensions | 256 |
| Type of Latent states $z_t$ | Gaussian |
| batch size | 16 |
| horizon | 64 |
| **Critic** | |
| Horizon | 12 |
| learning rate | $3e^{-5}$ |
| TD $\lambda^{\text{reward}}$ | 0.9 |
| discount $\gamma$ | 0.99 |
| number of hidden layers | 2 |
| number of hidden units | 512 |
| Polyak factor | 0.99 |
| **Actor** | |
| Horizon | 4 |
| learning rate | $3e^{-5}$ |
| number of hidden layers | 2 |
| number of hidden units | 512 |

Table 2: Hyperparameters for EquiDreamer

