# OpenReview forum: "Accelerating Model-Based Reinforcement Learning Using Equivariance"
_ICLR.cc/2026/Conference — Submitted to ICLR 2026_

### Official Review · Reviewer_bvVt · 2025-10-27

**Soundness:** 1
**Presentation:** 2
**Contribution:** 1
**Rating:** 2
**Confidence:** 5

**Summary:**

The paper studies equivariance in model-based reinforcement learning under POMDPs. Its main innovation is an equivariant Recurrent State Space model that reconstructs equivariant feature embeddings, rather than raw images, from a pretrained encoder using frame averaging. The method is evaluated on four tasks from the DeepMind Control Suite and compared against DreamerV3 and DrQ-v2.

**Strengths:**

* The problem is well-motivated; leveraging environment symmetries can meaningfully improve learning efficiency.
* The approach of reconstructing equivariant feature embeddings instead of raw observations is conceptually interesting.

**Weaknesses:**

1. The method assumes prior knowledge of the symmetry groups. It would be valuable to explore whether the approach could be extended to learn these symmetries automatically, as in [2,3].

2. Additionally, the method cannot handle approximate symmetries, and this limitation is not discussed. Addressing both these points could significantly strengthen the contribution and impact of the work.

3. I identified potential issues with Proposition 1 and its proof that questions soundness of the paper: Proposition 1 is derived from Equation (3), which in turn is taken from [1], an unpublished paper. Given the central role of this equation, the authors should better justify its validity rather than assuming it holds, especially since [1] has not undergone peer review and appears to be an informal note.

4. The proof relies on two strong and arguably unrealistic assumptions: (1) the dynamics model perfectly approximates the true transition dynamics, and (2) the symmetries are exact and perfectly generalizable. These assumptions are almost impossible to hold in neural network-based models and, importantly, prevent the method from being applicable to environments with approximate symmetries.

5. Results are reported on only four DMC environments out of more than twenty available. The selected tasks are all 2D with small action spaces, despite the method being motivated as generally applicable to 3D symmetries. This raises questions about scalability to more complex environments.

6. The reported baseline results, particularly for DrQ-v2, are inconsistent with the original paper, where the method performs much better. Here, DrQ-v2 fails to learn in nearly all environments, suggesting potential issues with implementation or hyperparameter choices. If different settings were used, the authors should clarify this. In any case, the authors need to provide a proper representation of the baseline, particularly if the baseline is well-established in the community and has a high reproducibility.

7. The paper is missing various baselines from equivariant representation learning algorithms. The method is only compared against Dreamer and DRQ-v2. At least one baseline from each of the following families should be included:
    * Model-free equivariant methods: Deep homomorphic policy gradient [2], or EQR [3].
    * Model-based equivariant methods:  EDGI [4], equivariant MuZero [5], SEN [7], or [8].

8. The contributions appear incremental compared to prior work on equivariant model-based RL. The paper does not cite or compare against some key methods in this area, such as [4] and [5].

Given the above methodological, theoretical, and empirical limitations, as well as concerns regarding the validity of the main proposition, I do not believe the paper meets the standard of this venue. Nonetheless, this is a promising research direction, and I encourage the authors to address these issues in future revisions.

### References
[1] Jiang, Nan. "A note on loss functions and error compounding in model-based reinforcement learning." arXiv preprint arXiv:2404.09946 (2024).

[2] Rezaei-Shoshtari, Sahand, et al. "Continuous mdp homomorphisms and homomorphic policy gradient." Advances in Neural Information Processing Systems 35 (2022): 20189-20204.

[3] Mondal, Arnab Kumar, et al. "Eqr: Equivariant representations for data-efficient reinforcement learning." International Conference on Machine Learning. PMLR, 2022.

[4] Brehmer, Johann, et al. "Edgi: Equivariant diffusion for planning with embodied agents." Advances in Neural Information Processing Systems 36 (2023): 63818-63834.

[5] Deac, Andreea, Théophane Weber, and George Papamakarios. "Equivariant MuZero." arXiv preprint arXiv:2302.04798 (2023).

[6] Park, Jung Yeon, et al. "Learning Symmetric Embeddings for Equivariant World Models." International Conference on Machine Learning. 2022.

[7] Zhao, Linfeng, et al. "Equivariant action sampling for reinforcement learning and planning." arXiv preprint arXiv:2412.12237 (2024).

**Questions:**

1. How are you imposing the value function to be invariant?
2. How does the method work with approximate symmetries in the environment?

---

> ### Author Response · Authors · 2025-11-27
>
> We would like to thank the reviewer for their time and feedback. Below we discuss the points raised in the review.
>
> > The method assumes prior knowledge of the symmetry groups. It would be valuable to explore whether the approach could be extended to learn these symmetries automatically, as in [2,3].
>
> While we agree that symmetry discovery is an interesting line of research. The assumption of prior knowledge of existing symmetries in the environment is realistic and reasonable, and has been shown to be effective in previous work, especially in robotics [1,2,3,4]. We consider the question of discovering equivariances to be outside the scope of our paper method and do not consider it as a weakness for our method.
>
> > Additionally, the method cannot handle approximate symmetries, and this limitation is not discussed. Addressing both these points could significantly strengthen the contribution and impact of the work.
>
> We disagree with the reviewer on this point. The symmetries in the DMC environments are approximate due to the checkerboard at the bottom of the observations and the stars in the background, leading to the image observations not being perfectly symmetric [5]. Further, we have added robot manipulation experiments where we have SO(2) equivariance and use the discretized C4 rotational group [1,2,3], and we find that even with a discretization of the group, despite not fully capturing the SO(2) equivariance. Leveraging equivariance leads to a strong improvement in performance. Further, previous work has shown that equivariant neural networks can lead to improvements to the method even when the symmetry in the observations do not have perfect symmetry [5].
>
> > I identified potential issues with Proposition 1 and its proof that questions soundness of the paper: Proposition 1 is derived from Equation (3), which in turn is taken from [1], an unpublished paper. Given the central role of this equation, the authors should better justify its validity rather than assuming it holds, especially since [1] has not undergone peer review and appears to be an informal note.
>
> We have added more citations to the simulation lemma.
>
> > The proof relies on two strong and arguably unrealistic assumptions: (1) the dynamics model perfectly approximates the true transition dynamics, and (2) the symmetries are exact and perfectly generalizable. These assumptions are almost impossible to hold in neural network-based models and, importantly, prevent the method from being applicable to environments with approximate symmetries.
>
> In many proofs of deep RL methods, ideal assumptions are made to gain intuition for the method's impact, and guide its design; we believe this was no different in our proof. Nevertheless, we removed the assumption of perfect dynamics and instead assumed bounded error for the transition model within the support of the training data, which is a reasonable assumption for neural networks given their local generalization ability. We also removed the assumption of perfect symmetry and now assume bounded error in symmetric configurations, given that previous work has shown that, under imperfect equivariance, equivariant NNs still lead to improvements in performance [5].
>
> Results are reported on only four DMC environments out of more than twenty available. The selected tasks are all 2D with small action spaces, despite the method being motivated as generally applicable to 3D symmetries. This raises questions about scalability to more complex environments.
>
> In addition to the DMC domains, we have added robot manipulation environments from bulletarm [6] showing the effectiveness of our approach in robot manipulation tasks. We only show SO(2) equivariance similar to [1,2,3], and leave experiments for SE(3) equivariance to future work.
>
> > The reported baseline results, particularly for DrQ-v2, are inconsistent with the original paper, where the method performs much better. Here, DrQ-v2 fails to learn in nearly all environments, suggesting potential issues with implementation or hyperparameter choices. If different settings were used, the authors should clarify this. In any case, the authors need to provide a proper representation of the baseline, particularly if the baseline is well-established in the community and has a high reproducibility.
>
> We used the official implementation with official HPs. As we are mostly interested in the sample numbers realistic for robot learning. We ran our experiments till 1e5-2.5e5 environment steps. We checked the results of DrQ-v2 in Dreamerv3[7] and in DrQ-v2[8], and they are in line with what we see in our results.

---

> > ### Author Response · Authors · 2025-11-27
> >
> > > The contributions appear incremental compared to prior work on equivariant model-based RL. The paper does not cite or compare against some key methods in this area, such as [4] and [5].
> >
> > We disagree with the reviewer, given that our method is the first paper that combines frame-averaging and reconstruction in the feature space to learn equivariant representation, as well as studying the effect of equivariance on the simulation lemma, and presenting an equivariant model-based RL method that can handle POMDPs. We find that our contributions are novel compared to previous work.
> >
> > Specifically, [9] is an offline RL method that does not handle online learning, where the agent needs to learn and explore online. Unfortunately, the method doesn’t explicitly say the number of demonstrations used but judging by those used in diffuser, their base method it’s >1000 demonstrations per task, which is a completely different setting to ours. Further, EDGI does not investigate learning equivariant representations for POMDPs, doesn’t study the effect of equivariance on the simulation lemma, and does not address questions like equivariant representation learning using reconstruction objectives. [10] It is not designed to handle continuous state/action spaces or to solve POMDPs; also, the paper has 3 citations and has not been published [11] EQR is a model-free method in which learning the transition model is used only to supervise the learning of equivariant representations; it thus doesn’t use model rollouts for learning, but rather to study the problem of learning equivariant representations that are unknown a priori. Further, EQR targets MDPs, whereas our method generalizes to POMDPs. In addition, the transition model in EQR is linear, which limits its expressiveness compared to learning a nonlinear transition function as in our method. We consider the paper to be orthogonal to our work. [12] The method is based on TD-MPC, which uses entropy-based planning. This makes enforcing equivariance much more difficult than learning an equivariant policy using equivariant NNs, which is the approach we take in our work. Further, the method doesn’t extend to POMDPs and doesn’t study the effect of equivariance on the simulation lemma. Finally, in our paper, we study how to learn equivariant stochastic latent-variable states that ensure variability in rollouts when using model data for training policies and actions. Unfortunately, the code is also not open-sourced, so we can’t compare our method to it. [13] SEN studies the case of learning representation learning, where the symmetry in the observation space might not be entirely known while the symmetry in the latent space is known. The paper is not a model-based RL paper and is obviously orthogonal to our work.
> >
> > > How are you imposing the value function to be invariant?
> >
> > We are implementing the value function as an invariant function; this essentially boils down to polling across the dimension of the different group elements in the feature map.
> >
> > > How does the method work with approximate symmetries in the environment?
> >
> > Given the results in [5], we can see that using equivariant neural networks helps make symmetric configurations in distribution for the learned functions, even for imperfect observations. We see that this can help with model-free RL methods as seen in [5]. Our results also suggest that this has also been the case for our approach.
> >
> > We hope our arguments will convince the reviewer to adjust the score for our paper.

---

> > > ### Author Response · Authors · 2025-11-27
> > >
> > > [1] Wang, Dian, Robin Walters, and Robert Platt. "$\mathrm {SO}(2) $-Equivariant Reinforcement Learning." International Conference on Learning Representations.
> > >
> > > [2]Nguyen, Hai Huu, et al. "Equivariant reinforcement learning under partial observability." Conference on Robot Learning. PMLR, 2023.
> > >
> > > [3]Wang, Dian, et al. "On-Robot Learning With Equivariant Models." 6th Annual Conference on Robot Learning.
> > >
> > > [4] Wang D, Hart S, Surovik D, et al. Equivariant Diffusion Policy[C]//8th Annual Conference on Robot Learning.
> > >
> > > [5]  Wang, Dian, et al. "The Surprising Effectiveness of Equivariant Models in Domains with Latent Symmetry." The Eleventh International Conference on Learning Representations.
> > >
> > > [6] Wang, Dian, et al. "Bulletarm: An open-source robotic manipulation benchmark and learning framework." The International Symposium of Robotics Research. Cham: Springer Nature Switzerland, 2022.
> > >
> > > [7] Hafner, Danijar, et al. "Mastering diverse control tasks through world models." Nature (2025): 1-7.
> > >
> > > [8] Yarats, Denis, et al. "Mastering Visual Continuous Control: Improved Data-Augmented Reinforcement Learning." International Conference on Learning Representations.
> > >
> > > [9] Brehmer, Johann, et al. "Edgi: Equivariant diffusion for planning with embodied agents." Advances in Neural Information Processing Systems 36 (2023): 63818-63834.
> > >
> > > [10] Deac, Andreea, Théophane Weber, and George Papamakarios. "Equivariant MuZero." arXiv preprint arXiv:2302.04798 (2023).
> > >
> > > [11] Mondal, Arnab Kumar, et al. "Eqr: Equivariant representations for data-efficient reinforcement learning." International Conference on Machine Learning. PMLR, 2022.
> > >
> > > [12] Zhao, Linfeng, et al. "Equivariant action sampling for reinforcement learning and planning." arXiv preprint arXiv:2412.12237 (2024).
> > >
> > > [13] Park, Jung Yeon, et al. "Learning Symmetric Embeddings for Equivariant World Models." International Conference on Machine Learning. PMLR, 2022.

---

### Official Review · Reviewer_i4Nz · 2025-10-31

**Soundness:** 2
**Presentation:** 2
**Contribution:** 2
**Rating:** 4
**Confidence:** 3

**Summary:**

This paper proposes EQUIDREAMER, a model-based framework that incorporates equivariance into the latent dynamics model and policy learning for POMDPs. The method uses symmetries in the environment to improve sample efficiency and generalization, building on the Dreamer architecture by replacing image reconstruction with feature reconstruction and using equivariant neural networks. Experiments on visual control tasks demonstrate improved performance over DreamerV3 and DrQ-v2 baselines.

**Strengths:**

1. Integrating equivariance principles into a model-based RL framework is an interesting research area, as well as revealing its sample efficiency in many problems.
2. Experimental results show consistent improvements in sample efficiency on several continuous control tasks.

**Weaknesses:**

1. The overall novelty is limited, as the core idea of applying equivariance to RL has been explored in prior work (e.g, [1][2]), and the extension to model-based RL, while sensible, does not constitute a significant conceptual advance.

2. The claimed benefits of feature reconstruction over image reconstruction are not sufficiently analyzed from the equivariance component, making it difficult to attribute the gains to the proposed novelty.

3. The model structure is confusing, particularly the use of a unified parameterization $p_{\theta}$ for the transition, observation, and reward models. This convolutes the roles of distinct components and lacks a clear justification.

minor typos:
1. In line 149, it seems like the equation form is incorrect.
2. In line 151, it should be "In this paper,..."

[1] Mondal, A. K., Jain, V., Siddiqi, K., & Ravanbakhsh, S. (2022, June). Eqr: Equivariant representations for data-efficient reinforcement learning. In International Conference on Machine Learning (pp. 15908-15926). PMLR.

[2] Grimm, C., Barreto, A., Singh, S., & Silver, D. (2020). The value equivalence principle for model-based reinforcement learning. Advances in neural information processing systems, 33, 5541-5552.

**Questions:**

1. How does the method scale to more complex symmetry groups or real-world tasks where symmetries merely exist?
2. Were there any environments or symmetry conditions where the equivariant model failed to improve upon the baseline?

---

> ### Author Response · Authors · 2025-11-27
>
> We appreciate the reviewer’s time and feedback. In the following, we address the concerns and questions raised.
>
> > The overall novelty is limited, as the core idea of applying equivariance to RL has been explored in prior work (e.g, [1][2]), and the extension to model-based RL, while sensible, does not constitute a significant conceptual advance.
>
> We respectfully disagree with the reviewer. Our method provides multiple innovations to combine equivariant neural networks and Dreamer. Primarily devising an architecture to supervise the dreamer model while maintaining equivariance. The original Dreamer papers rely on image reconstruction, which, in our case, would require learning 2x2 latent tensors to encode spatial equivariance, which is computationallyionally and memory expensive. To sidestep this issue, we employ pretrained ResNet encoders, which we also use as a fixed training target for our method. To maintain equivariance, we use frame averaging. As far as we know, our method is the first to investigate learning world models by reconstructing in an equivariant feature space. Further, our method is the first equivariant MBRL to handle POMDPs, as illustrated in our robot manipulation experiments. Our paper investigated the effect of leveraging symmetry on the simulation lemma bound, and to the best of our knowledge, this hasn’t been previously investigated. Looking at [1]: EQR is a model-free method where learning the transition model is only used for supervising the learning of equivariant representations, i.e., the method thus doesn’t use model rollouts for learning, but rather to study the issue of learning equivariant representations that are unknown a priori. Further, EQR only targets MDPs, whereas our method generalizes to POMDPs as well. In addition, as far as we can tell, the transition model in EQR is linear, which limits its expressiveness compared to learning a nonlinear transition function as in our method. We consider EQR to be orthogonal to our work. [8] I have gone through the paper, and I would say that it provides a very nice motivation to the use of equivariance in model-based RL, [8] can be seen more as providing conceptual motivation to our method but it doesn’t address many of the algorithmic contributions made in our paper as also acknowledged in there conclusion section.
>
> >The claimed benefits of feature reconstruction over image reconstruction are not sufficiently analyzed from the equivariance component, making it difficult to attribute the gains to the proposed novelty.
>
> We have an ablation on that in Figure 6, which shows that equivariance improves the performance when using feature reconstruction, actually Dreamer+ underperforms Dreamerv3 in cartpole/acrobot/cup_catch. The benefit of feature reconstruction is mainly saving compute of reconstructing every pixel in the image regardless of relevance to the task, which can be prohibitively computationally expensive in the case of reconstructing high-resolution images. Recent methods like JEPA[1] has shown the effectiveness of learning representations in the feature space rather than in the image space. We also find that using reconstruction in the feature space makes our model lighter. As shown in Table 1 in Appendix D, Performance-wise, an encoder trained specifically for the task would perform better than a pretrained encoder; however, image reconstruction might be feasible for high-resolution observations.
>
> > The model structure is confusing, particularly the use of a unified parameterization for the transition, observation, and reward models. This convolutes the roles of distinct components and lacks a clear justification.
>
> Here, we have followed the same notation as in Dreamer and other model-based RL methods [2,3,4,5]. We considered using different symbols, but this might confuse readers accustomed to the common notations in dreamer-based models.

---

> ### Author Response · Authors · 2025-11-27
>
> **Question:**
>
> > How does the method scale to more complex symmetry groups or real-world tasks where symmetries merely exist?
>
> We are mostly concerned with robotic use-cases. We investigated the SO(2) equivariance similar to [9,10,11].
>
> > Were there any environments or symmetry conditions where the equivariant model failed to improve upon the baseline?
>
> I can imagine that in environments where the symmetry chosen for the model is detrimental, the task can be quite adverse to learning, and the agent might fail. This has been investigated in [6], where the authors showed imperfect symmetries and found that if the symmetry chosen in the model contradicts that in the MDP then the model might not learn well.
>
> We hope that we have addressed the reviewer’s concerns sufficiently and that they will consider raising the score.
>
> [1] Assran, Mahmoud, et al. "Self-supervised learning from images with a joint-embedding predictive architecture." Proceedings of the IEEE/CVF Conference on Computer Vision and Pattern Recognition. 2023.
>
> [2] Hafner, Danijar, et al. "Mastering diverse control tasks through world models." Nature (2025): 1-7.
>
> [3] Wu, Philipp, et al. "Daydreamer: World models for physical robot learning." Conference on robot learning. PMLR, 2023.
>
> [4] Hafner, Danijar, et al. "Deep hierarchical planning from pixels." Advances in Neural Information Processing Systems 35 (2022): 26091-26104.
>
> [5] Lambrechts, Gaspard, Adrien Bolland, and Damien Ernst. "Informed POMDP: Leveraging Additional Information in Model-Based RL." Reinforcement Learning Conference.
>
> [6] Wang, Dian, et al. "The Surprising Effectiveness of Equivariant Models in Domains with Latent Symmetry." The Eleventh International Conference on Learning Representations.
>
> [7] Mondal, Arnab Kumar, et al. "Eqr: Equivariant representations for data-efficient reinforcement learning." International Conference on Machine Learning. PMLR, 2022.
>
> [8] Grimm, Christopher, et al. "The value equivalence principle for model-based reinforcement learning." Advances in neural information processing systems 33 (2020): 5541-5552.
>
> [9] Wang, Dian, Robin Walters, and Robert Platt. "$\mathrm {SO}(2) $-Equivariant Reinforcement Learning." International Conference on Learning Representations.
>
> [10]Nguyen, Hai Huu, et al. "Equivariant reinforcement learning under partial observability." Conference on Robot Learning. PMLR, 2023.
>
> [11]Wang, Dian, et al. "On-Robot Learning With Equivariant Models." 6th Annual Conference on Robot Learning.

---

### Official Review · Reviewer_vinT · 2025-10-31

**Soundness:** 3
**Presentation:** 3
**Contribution:** 2
**Rating:** 6
**Confidence:** 3

**Summary:**

This paper proposes a equivariant MBRL method to capture symmetries in specific POMDP domains. It helps with generalization to unseen equivariant states during training thus improving sample efficiency and meanwhile incorporates physical information. The proposed equivariant model-based RL method, EquiDreamer, shows higher sample efficiency compared to DreamerV3, demonstrating the effectiveness of method proposed.

**Strengths:**

1. It's an important research subject to discover symmetric patterns during training to reduce abundant exploration and boost sample efficiency.

2. The effectiveness of components of method proposed is clearly supported and discussed with empirical evidence.  The idea of reconstructing in the feature space instead of the original pixel space shows advantage on complex tasks like reacher-hard.

**Weaknesses:**

1. Visualization of the symmetries discovered could be presented to help understanding the method proposed. It remains somehow vague that whether EquiDreamer actually captured the equivariance between distinct state.

2. There are only empirical results on five tasks in DMC, which weakens the evidence for the effectiveness and generalizability of the method proposed. Results in other continuous control domains like Robodesk, Meta-world which also seem to have inherent symmetries would be preferred.

**Questions:**

1. Despite DMC, can you list more benchmarks or domains that inherit the equivariant feature?

---

> ### Author Response · Authors · 2025-11-27
>
> We would like to thank the reviewer for his time and feedback. We have tried to address the concerns in our rebuttal and revised version. Here we address the reviewer’s individual comments.
> > Visualization of the symmetries discovered could be presented to help understanding the method proposed
>
> In our method, we do not discover the symmetries; rather, we assume domain knowledge of the environment’s symmetry. This assumption is realistic in robotic manipulation use-cases where the agent’s policy needs to be equivariant to SO(2) or SE(3) transformations to the manipulated object. In our paper, we only experiment on SO(2) equivariance similar to [1,2,3].
>
> > There are only empirical results on five tasks in DMC, which weakens the evidence for the effectiveness and generalizability of the method proposed. Results in other continuous control domains like Robodesk, Meta-world which also seem to have inherent symmetries would be preferred.
>
> We’ve added four robot manipulation environments from the bulletarm simulation [3] that have been used in [1,2,3], including a partially observable task where the agent needs to gather information on the scene to achieve the task. We find that leveraging equivariance, we get strong performance.
>
> **Questions:**
>
> Despite DMC, can you list more benchmarks or domains that inherit the equivariant feature?
> We’ve added four robot manipulation environments from the bulletarm simulation [3] that have been used in [1,2,3], including a partially observable task where the agent needs to gather information on the scene to achieve the task. We find that leveraging equivariance yields strong performance.
>
> We hope that we have addressed the reviewer's concerns about our paper and hope that the reviewer will consider raising the score as a result.
>
> [1] Wang, Dian, Robin Walters, and Robert Platt. "$\mathrm {SO}(2) $-Equivariant Reinforcement Learning." International Conference on Learning Representations.
>
> [2]Nguyen, Hai Huu, et al. "Equivariant reinforcement learning under partial observability." Conference on Robot Learning. PMLR, 2023.
>
> [3]Wang, Dian, et al. "On-Robot Learning With Equivariant Models." 6th Annual Conference on Robot Learning.

---

### Author Response · Authors · 2025-11-27

We thank the reviewers for their time and feedback. We have done our best to incorporate the suggestions in our manuscript and reply to the comments in detail. Here, we highlight our changes to the manuscript and highlight the novelty of our method and discuss our experiments. We reply to each individual review in more detail in our comments to each review. Here we address some general points

First, we clarify that our method does not attempt to discover unknown symmetries; rather, we leverage known and symmetries commonly present in robotic manipulation tasks. This design choice is consistent with prior work [1,2,3] and allows us to focus on how to effectively integrate equivariance into sota model-based RL algorithms. To strengthen our empirical validation, we expanded our experiments beyond DMC to include four robot manipulation environments from BulletArm, including a partially observable setting. These results confirm that incorporating equivariance into world models yields consistent and significant performance gains, even under approximate symmetries.

Second, we highlight the key algorithmic novelties of our approach. Our method introduces an equivariant world model for Dreamer-style MBRL that reconstructs in an equivariant feature space rather than pixel space—avoiding the computational burden of image reconstruction while preserving equivariant structure. We also combine pretrained encoders with frame averaging to benefit from powerful pretrained encoders and provide features as stable training targets to supervise the latent dynamics model. Combined with leveraging the deterministic/stochastic latent states, enabling the first equivariant MBRL system capable of handling POMDPs. Additionally, we provide, to the best of our knowledge, the first analysis of how symmetry affects the simulation lemma in model-based RL. In response to the reviewer's concerns, we refined the proof to rely solely on bounded modeling error and approximate symmetries, aligning it with realistic neural network behavior.

We hope that the additional experiments and modifications to the manuscript have addressed the authors' concerns, and we are happy to clarify any remaining issues.

Here is a summary of the changes in the manuscript. The added parts in the manuscript are highlighted in blue.

**Change log**

- We have added robot manipulation results, that use SO(2) group equivariance similar to [1,2,3].
- Change simulation lemma citation.
- Adapted our proof to avoid assuming perfect symmetries and perfect dynamics in visited states.
- Updated related work.
- Fix minor mistakes in the text.

[1] Wang, Dian, Robin Walters, and Robert Platt. "$\mathrm {SO}(2) $-Equivariant Reinforcement Learning." International Conference on Learning Representations.

[2]Nguyen, Hai Huu, et al. "Equivariant reinforcement learning under partial observability." Conference on Robot Learning. PMLR, 2023.

[3]Wang, Dian, et al. "On-Robot Learning With Equivariant Models." 6th Annual Conference on Robot Learning.

---

### Meta-Review · Area_Chair_brZ4 · 2025-12-30

**Summary:**

The reviewers appreciated the paper’s solid technical execution, clear motivation, and consistent empirical improvements from incorporating equivariant structure into a model-based RL framework. However, three main concerns informed the suggested decision: (1) the contribution was widely viewed as incremental, with equivariance being a well-studied concept and its integration into Dreamer seen as a natural extension rather than a conceptual advance; (2) the method relies on the assumption of known and exact symmetries, which limits applicability and leaves robustness under approximate or unknown symmetries unaddressed; and (3) although additional ablations were provided, questions remain about how cleanly the performance gains can be attributed to equivariance itself as opposed to other design choices such as feature reconstruction.

**Reviewer Concerns:**

The rebuttal and additional ablation experiments reasonably addressed concern (3) by better isolating the effect of equivariance from other design choices, thereby strengthening the empirical evidence for the claimed performance gains. However, concerns (1) and (2) remain outstanding: the contribution is still fundamentally incremental, and the reliance on known symmetries continues to limit the scope and impact of the work. In this regard, AC largely agrees with the reviewers that the limited novelty and the symmetry assumption constrain the overall contribution of the paper.

**Reviewer Scores:**

The fundamental limitations regarding novelty and the reliance on known symmetries suggest that any score changes would likely be small. Overall, the discussion would not be expected to lead to substantial score increases across reviewers.

---

### Decision · Program_Chairs · 2026-01-26

Reject